# Augmented K_Ca_2.3 Channel Feedback Regulation of Oxytocin Stimulated Uterine Strips from Nonpregnant Mice

**DOI:** 10.3390/ijms222413585

**Published:** 2021-12-18

**Authors:** Megan Zak, Bri Kestler, Trudy Cornwell, Mark S. Taylor

**Affiliations:** Department of Physiology and Cell Biology, University of South Alabama College of Medicine, Mobile, AL 36688, USA; Megan.Zak@hsc.utah.edu (M.Z.); brikestler@southalabama.edu (B.K.); tcornwell@retired.southalabama.edu (T.C.)

**Keywords:** uterine contraction, myometrium, tocolytic, K_Ca_2.3, CyPPA, internal calcium store

## Abstract

Uterine contractions prior to 37 weeks gestation can result in preterm labor with significant risk to the infant. Current tocolytic therapies aimed at suppressing premature uterine contractions are largely ineffective and cause serious side effects. Calcium (Ca^2+^) dependent contractions of uterine smooth muscle are physiologically limited by the opening of membrane potassium (K^+^) channels. Exploiting such inherent negative feedback mechanisms may offer new strategies to delay labor and reduce risk. Positive modulation of small conductance Ca^2+^-activated K^+^ (K_Ca_2.3) channels with cyclohexyl-[2-(3,5-dimethyl-pyrazol-1-yl)-6-methyl-pyrimidin-4-yl]-amine (CyPPA), effectively decreases uterine contractions. This study investigates whether the receptor agonist oxytocin might solicit K_Ca_2.3 channel feedback that facilitates CyPPA suppression of uterine contractions. Using isometric force myography, we found that spontaneous phasic contractions of myometrial tissue from nonpregnant mice were suppressed by CyPPA and, in the presence of CyPPA, oxytocin failed to augment contractions. In tissues exposed to oxytocin, depletion of internal Ca^2+^ stores with cyclopiazonic acid (CPA) impaired CyPPA relaxation, whereas blockade of nonselective cation channels (NSCC) using gadolinium (Gd^3+^) had no significant effect. Immunofluorescence revealed close proximity of K_Ca_2.3 channels and ER inositol trisphosphate receptors (IP_3_Rs) within myometrial smooth muscle cells. The findings suggest internal Ca^2+^ stores play a role in K_Ca_2.3-dependent feedback control of uterine contraction and offer new insights for tocolytic therapies.

## 1. Introduction

The myometrium undergoes dynamic modifications during gestation to promote uterine quiescence for healthy fetal development. Disruption of these changes can lead to coordinated, phasic uterine contractions before a full 37-week gestation and result in preterm labor. Preterm labor and premature birth increase infant morbidity and mortality as well as health care cost [1]. Early onset contractions may be induced by a variety of factors, including infection, hormonal dysregulation, placental rupture, and maternal-fetal stress, making targeted treatment difficult [2]. Phasic uterine contractions result from the influx of extracellular calcium (Ca^2+^) through voltage-gated Ca^2+^ channels (VGCCs) [3]. These periodic Ca^2+^ spikes spread throughout the myometrium to produce coordinated, productive contractions during labor. Commonly used tocolytic therapies for reducing preterm uterine contractions include magnesium sulfate and nifedipine, both of which work by reducing Ca^2+^ influx [4,5]. While these therapies can delay delivery by approximately 48 h to allow administration of corticosteroids to support fetal lung maturity, they fail to prevent preterm labor [6]. The danger of these treatments lies in their adverse side effects. Because VGCCs channels are expressed in a variety of tissues in the body [7], acute inhibition can result in serious deleterious effects, including maternal hypotension, pulmonary edema and cardiac arrest [4,6]. Finding more targeted tocolytic therapies could offer safer treatment for mothers and reduce rates of infant morbidity and mortality.

Potassium channels play an important role in the regulation of uterine contractions, primarily by repolarizing contractile action potentials and reducing uterine excitability. This regulation involves calcium-activated potassium channels (K_Ca_), including a class of small conductance channels (K_Ca_2) that exhibit potent Ca^2+^ sensitivity. In particular, the K_Ca_2.3 channel subtype has been implicated as a key controller of uterine contractility and may be a promising target for attenuating premature uterine contractions [8,9,10]. These channels are expressed in the myometrium [8,11] and act as feedback controllers of uterine contraction by opening in response to rising intracellular Ca^2+^ concentrations ([Ca^2+^]_i_) [12]. The resulting K^+^ efflux hyperpolarizes the cell membrane potential, which then decreases additional Ca^2+^ influx through VGCCs. Previous studies from our laboratory, and others, have shown that uterine contractions are substantially depressed, and parturition significantly delayed in K_Ca_2.3 overexpressing mice [8,13]. In fact, K_Ca_2.3 overexpression has been shown to prevent preterm labor in mice [10]. These effects can be mimicked by cyclohexyl-[2-(3,5-dimethyl-pyrazol-1-yl)-6-methyl-pyrimidin-4-yl]-amine (CyPPA), a positive modulator of K_Ca_2.3 channels [14] that acts to sensitize Ca^2+^-dependent K_Ca_2.3 channel opening and ultimately relax myometrial contraction. CyPPA was found to significantly reduce phasic uterine contractions in both non-pregnant and pregnant mice [9] as well as human myometrial tissue [15], suggesting the utility of this approach for future tocolytic therapy.

The peptide hormone oxytocin plays an important role in stimulating and coordinating uterine contractions at term. Although the uterus contracts primarily through regenerative pacemaker action potentials spreading through the myometrium, increased oxytocin release from the posterior pituitary in the later stages of gestation leads to higher circulating oxytocin levels and enhanced uterine contractions. During laboring contractions, binding of oxytocin to Gq-protein coupled receptors (GPCR) on the myometrium augments uterine contractility by soliciting additional increases in [Ca^2+^]_i_ [16]. This mechanism involves GPCR-mediated elevation of intracellular inositol trisphosphate (IP_3_), which binds to IP_3_ receptors (IP_3_R) on the endoplasmic reticulum (ER) membrane to induce release of intracellular Ca^2+^ stores into the cytosol. It has become clear in recent years that spatially localized Ca^2+^ release from internal stores can target plasma membrane K_Ca_ channels and exert direct feedback regulation in various tissues [17,18,19]. Notably, these restricted signals can occur without altering global cellular [Ca^2+^]_i_. In vascular smooth muscle, this feedback regulation by the endoplasmic reticulum is an important controller of arterial tone [17,20], ultimately regulating blood pressure and flow.

In this study, we assess the capacity of CyPPA to suppress uterine contractions stimulated by oxytocin and ask whether internal Ca^2+^ stores play a role in facilitating CyPPA relaxation. Our findings suggest that positive modulation of K_Ca_2.3 channels can effectively block oxytocin-mediated contractions and that internal Ca^2+^ stores contribute to this enhanced feedback response.

## 2. Results

### 2.1. CyPPA Prevents Oxytocin Potentiation of Uterine Contractions

We used isometric force myography to assess the impact of positive K_Ca_2.3 channel modulation on oxytocin-induced contractions of mouse uterine strips. In control strips, oxytocin (0.03–1 µM) increased phasic contractions (Figure 1). These responses typically involved a transient increase in frequency and tone (that waned with time) as well as a modest increases in amplitude. Pretreatment with 10 µM CyPPA blunted spontaneous contractions and prevented oxytocin-induced augmentation of force. These effects were most evident in contraction amplitude and AUC, while contraction frequency was not significantly different. Notably, in the presence of CyPPA, contractions remained muted, even at higher concentrations of oxytocin. Subsequent addition of the K_Ca_2 channel blocker, apamin, restored phasic contractions, supporting the principal role of K_Ca_2.3 channel activity in the suppression of contractions. 

### 2.2. Inhibition of SERCA Impairs CyPPA Relaxation

We considered that the strong CyPPA inhibition of uterine contractions in the presence of oxytocin might involve enhanced internal Ca^2+^ store release augmenting K_Ca_2.3 feedback. In order to assess whether internal Ca^2+^ stores contribute to K_Ca_2.3 mediated feedback suppression of uterine contractions, the intracellular Ca^2+^ stores were depleted by blocking the sarcoplasmic endoplasmic reticulum Ca^2+^ ATPase (SERCA) with 10 µM CPA before adding 100 nM oxytocin. Blocking reuptake allows Ca^2+^ to leak out of the ER, unloading the internal stores. We found that CPA pretreatment increased uterine contraction (Figure 2A,B) and significantly reduced CyPPA suppression of phasic uterine contractions compared to vehicle (Figure 2C). This effect was most apparent in contraction amplitude and AUC, particularly at CyPPA concentrations of 10 and 30 µM. These data support a role for the internal Ca^2+^ stores in the K_Ca_2.3 feedback control of uterine contractions. 

### 2.3. Inhibition of NSCCs Does Not Alter CyPPA Relaxation

In addition to triggering internal store release, GPCR agonist stimulation can also increase Ca^2+^ influx from the extracellular environment. In particular, store-operated Ca^2+^ entry and second messenger mediated influx through plasma membrane nonselective cation channels (NSCCs) have been implicated in agonist responses [21]. To address the possible contribution of this Ca^2+^ source to the K_Ca_2.3 dependent feedback, we pretreated uterine strips with 30 µM Gd^3+^ to block NSCC influx. Pretreatment with Gd^3+^ did not alter uterine contractions (Figure 3A,B) and although Gd^3+^ seemed to modestly impair CyPPA suppression of uterine contractions, this apparent effect did not reach statistical significance for the parameters evaluated (Figure 3C). 

### 2.4. K_Ca_2.3 Channels and IP_3_Rs Exhibit Overlapping Punctate Expression Patterns in the Myometrium

The implication of our functional data is that agonist-induced release of Ca^2+^ from the ER may provide preferential targeting of K_Ca_2.3 channels to solicit negative feedback control of membrane potential and uterine contractions. This could be similar to the mechanism previously reported in the vascular endothelium, whereby IP_3_Rs on the ER membrane reside in close proximity to plasma membrane K_Ca_ channels and elicit membrane hyperpolarization during GPCR stimulation [19]. To investigate the possibility of preferential localization of IP_3_R with K_Ca_2.3 channels within myometrial smooth muscle, thin sections of myometrium were removed from uterine strips and probed for K_Ca_2.3 and IP_3_R expression via immunofluorescence. Immunostaining revealed densities and puncta of IP_3_R and K_Ca_2.3 channels within longitudinal myometrial smooth muscle (Figure 4). Distinct overlapping densities of IP_3_Rs and K_Ca_2.3 channels suggests substantial spatial clustering of these channels within confined cellular spaces (1–5 µm).

## 3. Discussion

K_Ca_2.3 channels play an important role in the negative feedback control of uterine contractions. While amplification of this feedback by the positive K_Ca_2.3 modulator, CyPPA, has been described, its impact in the presence of the pro-contractile hormone oxytocin has not been fully elucidated. The current study shows that CyPPA effectively prevents oxytocin augmentation of contractions in mouse uterus. In fact, our findings suggest oxytocin may facilitate the anti-contractile mechanism of CyPPA by promoting release of internal Ca^2+^ stores in close proximity to membrane K_Ca_2.3 channels, further amplifying the feedback response. Therefore, the tocolytic impact of positive K_Ca_2.3 channel modulators may be serendipitously augmented under laboring conditions when endogenous pro-contractile stimuli are elevated. This seemingly paradoxical effect of oxytocin provides useful insight for future tocolytic approaches. It suggests that strategies designed to exploit or amplify Ca^2+^ feedback may be particularly effective in opposing agonist-facilitated laboring contractions. 

Blocking SERCA in myometrial smooth muscle allowed us to remove the internal ER Ca^2+^ store and, thereby, test its role in uterine contractions, including feedback control. Interestingly, blocking ER Ca^2+^ uptake alone caused an increase in uterine contraction. This is consistent with previous observations [8,22] and suggests a possible inherent negative feedback role of internal stores in the control of uterine contractility. As a positive modulator, CyPPA sensitizes the Ca^2+^-dependent opening of K_Ca_2.3 channels, essentially amplifying negative feedback. We surmised that depleting internal stores prior to oxytocin exposure would reduce the Ca^2+^ released in the proximity of K_Ca_2.3 channels, thereby substantially limiting feedback and the impact of CYPPA. Indeed, we found that CyPPA responses are decreased after internal store depletion, suggesting internal stores are involved in K_Ca_2.3 dependent feedback. It is possible that accumulation of cytosolic Ca^2+^ following SERCA blockade contributed to this effect through global [Ca^2+^]_i_ rise, but no net increase in sustained contractile tone was noted and increased [Ca^2+^]_i_ alone would be expected to increase CyPPA suppression of VGCC-dependent phasic contraction. We also considered that membrane Ca^2+^-permeable NSCCs channels may allow Ca^2+^ influx, particularly under agonist-stimulated conditions. This entry may occur through various transient receptor potential (TRP) channels, including members of the TRPC and TRPV families previously described in the myometrium [23,24]. Here, we used Gd^3+^ to block entry through a broad range of NSCCs. Although our findings did not clearly implicate NSCCs in K_Ca_2.3 feedback control of uterine function, these channels might play an important role under certain conditions. For instance, increasing uterine stretch or altered channel expression or distribution during pregnancy [24] could expand their effective coupling with K_Ca_ channels and increase their feedback influence. Indeed, we and others have reported coupling of TRPV4 Ca^2+^ influx and K_Ca_ channels [25,26,27], and recent findings suggest stimulation of membrane TRPV4 channels can elicit strong suppression of uterine contractions during pregnancy [28]. Future studies assessing concordant feedback mechanisms acting in unison during pregnancy will be particularly insightful. 

Our current findings highlight the internal Ca^2+^ store as a potential negative regulator of uterine contraction through K_Ca_ channels. This scenario may be analogous to mechanisms previously described in blood vessels. In vascular smooth muscle, release of Ca^2+^ sparks from the ER, located just under the plasma membrane, solicit large-conductance K_Ca_-dependent hyperpolarization and promote vasodilation [17,29]. Moreover, in the vascular endothelium, GPCR agonists tune vasodilation by stimulating Ca^2+^ transients (Ca^2+^ puffs or pulsars) from clusters of ER IP_3_Rs that target nearby membrane small conductance K_Ca_ channels to produce endothelium-derived hyperpolarization of the vascular wall [19]. This functional coupling involves very close association of IP_3_Rs and K_Ca_ channels within distinct endothelial cell microdomains. Here we show a very similar arrangement in the uterus with distinct juxtaposing densities of IP_3_Rs and K_Ca_2.3 channels evident within the myometrium. This arrangement may allow a portion of GPCR stimulated Ca^2+^ release to be preferentially targeted to K^+^-dependent hyperpolarization and relaxation of the myometrium (Figure 5). In this respect, agonists such as oxytocin can recruit an inherent negative feedback mechanism to tune contractions, and CyPPA amplifies this effect by increasing the Ca^2+^ sensitivity of the feedback system, leading to further reduction of Ca^2+^ influx through VGCCs. Indeed, we previously showed that CyPPA effectively truncates the phasic Ca^2+^ spikes that drive uterine contractions [9]. It should be noted that localized Ca^2+^ release events such as sparks or puffs have not been identified and characterized in the uterine myometrium. While the nature or existence of such events remains unclear, it is possible that localized ER Ca^2+^ release at the membrane coincides with the regular global Ca^2+^ spikes causing contraction (i.e., via VGCCs), obscuring measurement of a distinct local signal. It will be useful for future studies to determine the specific impact of amplified feedback on Ca^2+^ spike magnitude and duration in the myometrium. 

It should be noted that the current study focuses on the general impact of oxytocin rather than its impact under any specific physiologic (or exogenously augmented) condition. While oxytocin may exert impacts at picomolar to nanomolar circulating concentrations, effective concentrations and receptor distributions at its sites of action are variable and dynamic. We employed oxytocin at the high end of its effective functional range to ensure adequate opportunity to overcome CyPPA suppression of contractility and to fully assess its capacity in feedback amplification. The key indication from this work is that any elevation in oxytocin (physiologic or pharmacologic) may be surmountable by CyPPA and that increasing oxytocin may actually support CyPPA suppression of contraction. This will provide useful insight for future studies pursuing K_Ca_ feedback as a potential tocolytic strategy. 

A limitation of the current study was the focus on non-pregnant uterine tissues. Future studies should address implications at term and preterm. We expect that changing channel and receptor expression patterns, as well as tissue remodeling over the course of gestation, will impact feedback control. Alterations in physical parameters, such as stretch, may also tune feedback control (e.g., Ca^2+^ entry through stretch activated cation channels), perhaps expanding opportunities for intervention over the course of gestation. Based on the current findings, an extended evaluation of IP_3_Rs, particularly in pregnancy and during preterm labor, is warranted. Unfortunately, because available pharmacologic inhibitors/modulators of IP_3_Rs are notoriously nonselective and solicit various off-target effects [30], we were unable to explicitly address their functional role here. Additionally, while we assessed oxytocin and general IP_3_R expression as an initial evaluation, deeper study of GPCR stimulation (e.g., via prostaglandins) and discrimination of the specific IP_3_R isoform(s) involved in feedback regulation could help focus new interventional strategies. Ultimately, tocolytic approaches may require concurrent adjustment of multiple components for synergistic feedback control. Extension of the current studies should elucidate the broader capacity of K_Ca_ feedback mechanisms in the human myometrium and provide useful insight into the timing and targeting of preterm labor interventions, leading to safer and more effective tocolytic therapy. 

## 4. Materials and Methods

### 4.1. Tissue Preparation

Mice were euthanized with injection of sodium pentobarbital (100 μg/g) into the peritoneal cavity. The uteruses were harvested and placed in 4 °C physiological salt solution (PSS) buffer (containing in mM: 119 NaCl, 4.7 KCl, 1.2 MgSO_4_, 2.0 CaCl_2_, 23 NaHCO_3_, 10.5 glucose, 0.026 EDTA, 1.2 KH_2_PO_4_; pH 7.4). The uterine horns were then cleaned and dissected to isolate longitudinal muscle strips. All procedures were performed in accordance with the University of South Alabama Institutional Animal Care and Use Committee and the National Institutes of Health Guide on the Humane Treatment of Experimental Animals. 

### 4.2. Isometric Force Myography

Equal-length longitudinal strips of uterus were mounted on two pins of an isometric myograph (610M; DMT, Central Jutland, Denmark) and immersed in a bath of 5 mL of PSS buffer at 37 °C aerated with 95% O_2_–5% CO_2_. All strips were stretched to a baseline force of 1.5 mN, primed with 300 nM oxytocin, washed twice, and given time to establish a stable baseline with spontaneous contractions. Experiments were performed using two to four muscle strips simultaneously to allow for parallel assessment of treatment and control groups. Phasic contractions were recorded via Chart 7.0 and quantified offline. Stock solutions of oxytocin (O4375; Sigma-Aldridge, Saint Louis, MO, USA), CyPPA (2953; Tocris, Bristol, UK) and cyclopiazonic acid (1235; Tocris, Bristol, UK) were prepared in DMSO; gadolinium chloride (4741; Tocris, Bristol, UK) and apamin (ab120268; Abcam, Waltham, MA, USA) were dissolved in water. Average amplitude, average frequency, and total area under curve (AUC) were assessed over comparable 10-min intervals. To be considered an event for determination of amplitude and frequency, a force fluctuation must reach at least 10% of the maximal spontaneous peak contraction value. Assessment of AUC is made by determining the cumulative force relative to the baseline (i.e., starting value before the onset of a contraction) over the same time interval assessed for amplitude and frequency. AUC gives a measure of total force that includes not only the explicit phasic events but also small fluctuations and changes in tone.

### 4.3. Immunofluorescence

Longitudinal myometrial smooth muscle was isolated, laid flat, and pinned with tungsten wire to gel (Sylgard) blocks to prepare for immunostaining. Tissue was fixed with 4% formalin, washed with phosphate buffered saline (PBS), and permeabilized with 0.5% Triton. Bovine serum albumin (BSA) was added to block nonspecific binding, and tissue was incubated overnight with K_Ca_2.3 primary rabbit antibody (1:300; Alomone Labs Jerusalem, Israel) at 4 °C and IP_3_R primary goat antibody (1:250; MyBioSource, San Diego, CA, USA). Tissue was then washed with PBS, treated with AlexaFluor 568 anti-rabbit secondary antibodies (1:500) and Alexa 488 anti-goat secondary antibodies (1:250) (Thermo Fisher Scientific, Waltham, MA, USA) for one hour, and washed. Hoechst dye (1:2000) was added to tissue to stain nuclei. Preparations were viewed with a Nikon A1 confocal microscope using NIS Elements (Tokyo, Japan) software and then analyzed using ImageJ software.

### 4.4. Data Analysis

For all experiments, n = number of animals. Data are expressed as means ± SEM. Plotting and analysis were performed using GraphPad Prism 9 (San Diego, CA, USA). All data sets were assessed for normality using Shapiro–Wilk test. Unpaired *t*-test was used for comparison of two independent groups. Multiple-variable groups were compared using Two-way ANOVA with a Šídák’s multiple comparisons. *p* ≤ 0.05 was considered significant.

## Figures and Tables

**Figure 1 ijms-22-13585-f001:**
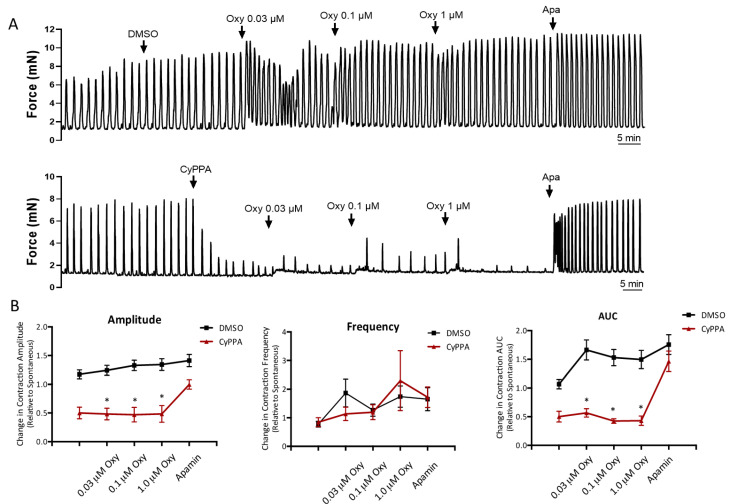
Oxytocin-induced contractions of isolated uterine strips in the presence or absence of CyPPA. (**A**) Panels show representative myography recordings of strips treated with either 10 μM CyPPA or 0.05% DMSO vehicle and subsequently stimulated with increasing concentrations (0.03–1 μM) of oxytocin. Finally, strips were exposed to the K_Ca_2 inhibitor apamin (0.6 μM). (**B**) Summary plots show cumulative effects of treatments on contraction amplitude, frequency, and area under curve (AUC). Asterisk (*) indicates *p* < 0.05 vs. DMSO; (n = 7).

**Figure 2 ijms-22-13585-f002:**
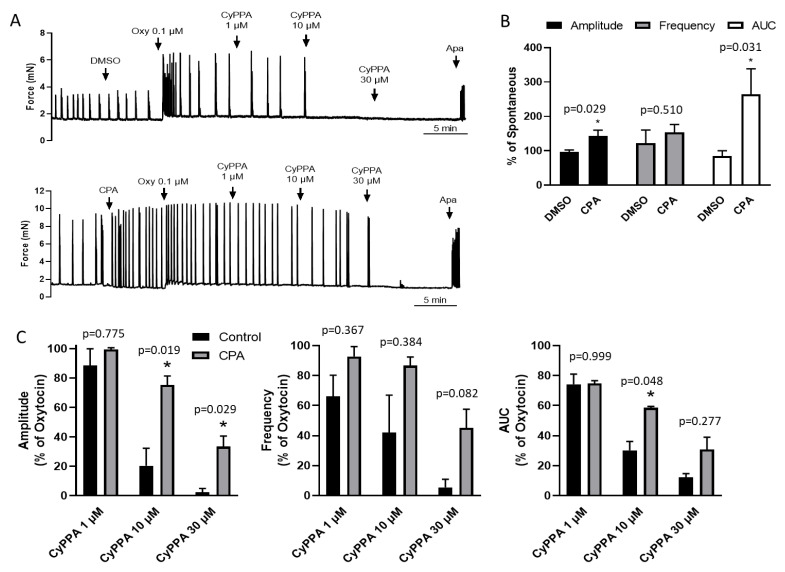
Effect of internal Ca^2+^ store depletion on CyPPA suppression of phasic uterine contractions. (**A**) Representative myography recording of uterine strips pretreated with CPA (10 µM) or DMSO vehicle (0.05%) before exposure to oxytocin and subsequent increasing concentrations of CyPPA. Finally, strips were exposed to the K_Ca_2 inhibitor apamin (0.6 µM). (**B**) Summary of CPA effects on contraction amplitude, frequency and AUC. (**C**) Summary of CyPPA effects on contraction amplitude, frequency and AUC under the conditions tested. Asterisk (*) indicates *p* < 0.05; (n = 6).

**Figure 3 ijms-22-13585-f003:**
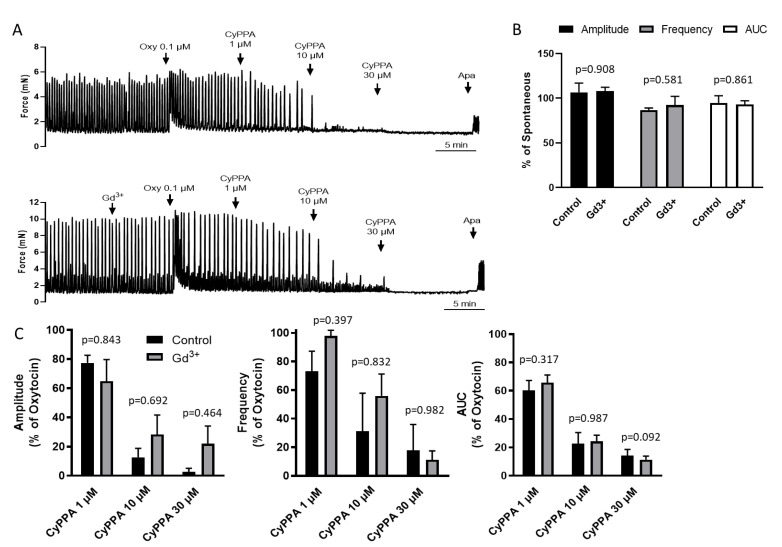
Effect of nonselective cation channel blockade on CyPPA suppression of phasic uterine contractions. (**A**) Representative myography recording of uterine strips pretreated with Gd^3+^ (30 µM) before exposure to oxytocin and subsequent increasing concentrations of CyPPA. Finally, strips were exposed to the K_Ca_2 inhibitor apamin (0.6 µM). (**B**) Summary of Gd^3+^ effects on contraction amplitude, frequency and AUC. (**C**) Summary of CyPPA effects on contraction amplitude, frequency and AUC under the conditions tested; (*n* = 6).

**Figure 4 ijms-22-13585-f004:**
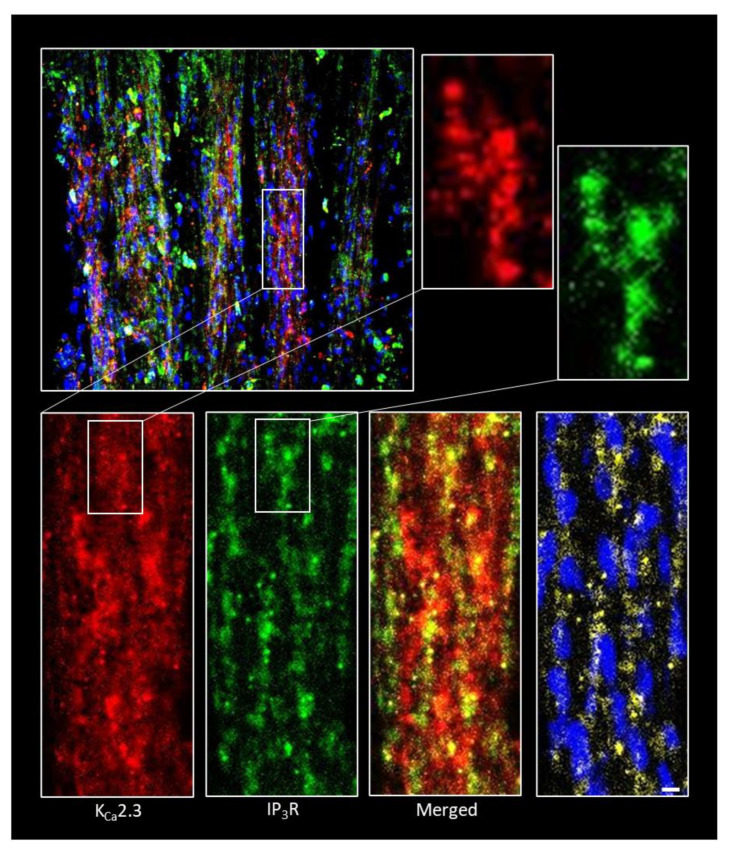
Expression of K_Ca_2.3 and IP_3_R in mouse myometrium. The upper left panel shows immunofluorescence staining of K_Ca_2.3 channels (red) and IP_3_Rs (green) in a longitudinal strip of mouse myometrium; cell nuclei are blue. The boxed region is expanded (bottom left) to show distinct K_Ca_2.3 and IP_3_R staining patterns; insets show zoom of the designated regions. The merged image reveals regions of overlapping K_Ca_2.3 and IP_3_R densities. The lower right panel highlights overlapping K_Ca_2.3 and IP_3_R-positive signals (yellow); cell nuclei are included for reference. Images are representative of tissue preparations from three animals. Scale bar 10 µm.

**Figure 5 ijms-22-13585-f005:**
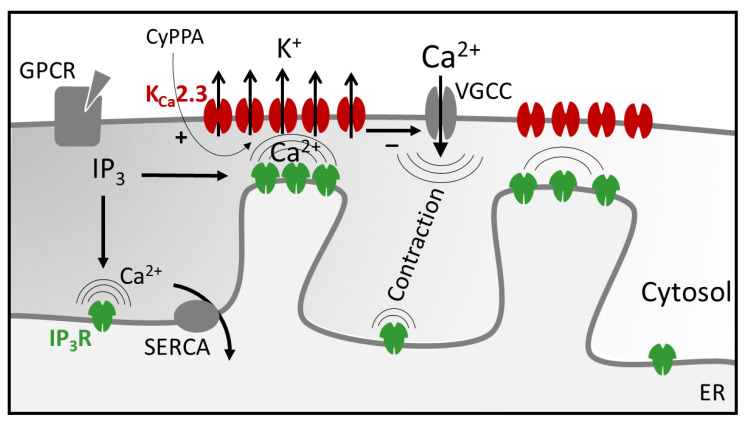
Schematic of proposed internal Ca^2+^ store feedback regulation of uterine contractions. Action potentials elicit phasic myometrial contractions by opening smooth muscle VGCCs. GPCR binding by oxytocin normally augments contractions via IP_3_-dependent release of internal (ER) Ca^2+^ stores. Concurrently, local Ca^2+^ release sites positioned near membrane K_Ca_2.3 channels solicit negative feedback control of VGCC influx, and limit overall contraction. Introduction of CyPPA increases the Ca^2+^ sensitivity of K_Ca_2.3 channels, thereby, potentiating K_Ca_2.3 feedback regulation. Heightened sensitization of internal store feedback may effectively block productive contractions in the presence of oxytocin.

## Data Availability

The data presented in this study are available on request from the corresponding author.

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
