# Peer review of "Augmented K_Ca_2.3 Channel Feedback Regulation of Oxytocin Stimulated Uterine Strips from Nonpregnant Mice"

_ijms, 2021, doi:10.3390/ijms222413585_

Round 1

Reviewer 1 Report

The authors have shown pharmacologically that KCa2.3 channels play an important role in the negative feedback control of uterine contractions. By focusing on oxytocin-induced uterine contractions in mice, the present study seems to be aimed at preventing preterm birth. Although the results are interesting and the purpose is clear, there are some points that need to be improved in terms of experimental conditions and interpretation of the results, which are described below.

1.The absence of a concentration-dependent increase in contraction by oxytocin needs to be discussed.

2.It should be discussed whether the oxytocin concentrations used in this study are similar to maternal plasma levels of oxytocin during physiological childbirth or intravenous oxytocin concentrations used for induction of labor or treatment of hypocontraction labor. This point is important in the discussion of whether the functional changes observed in this manuscript could account for the point of action in clinical practice.

3. Representative traces in Figure 2 should be replaced. According to the graphs in Fig. 2B,  CPA reduced the effects of CyPPA (10 uM and/or 30 uM) on contraction amplitude and AUC. However, in the Methods, the method of measuring the contraction amplitude and AUC is not described in detail, and it is unclear from what analysis the values in the graph were calculated. The effect of oxytocin at 0.1 uM in Fig. 2 (reduction of frequency) is quite different that in Fig.1A. This difference also needs to be explained.

4. Was the biphasic reaction always observed when 0.3 uM oxytocin was added in Fig. 1A?

5. According to the time-dependent effects of oxytocin, discuss how the time of drug administration may have affected the results of this study.

Reviewer 2 Report

The study by Zak and colleagues involves functional experiments (i.e. isometric force myography) employing various pharmacology challenges followed by immunofluorescence studies supporting the authors’ ideas of relevant protein clustering in non-pregnant mice. 

The functional data are reasonably convincing of the summary diagrammatic mechanism presented (Figure 5).

The study was performed in non-pregnant mice. The title and/or abstract of the study should clearly reflect this. 

Figure 1: DMSO, CyPPA and Apamin concentions should appear in the figure similar to those for oxytocin.  Bar charts in Figures 2 and 3 appear to demonstrate several cases of differences (e.g. Figure 2, panel B, frequency chart) that are not listed as significant. Can the authors provide the p value? It is likely that this is a feature of a limited “n number” group study, but would still be informative.  

A significant weakness of those study involves the lack of thorough mechanistic data. This is a problem particularly for whole-tissue/strip pharmacologic studies.  Attempts to demonstrate the mechanisms depicted in Figure 5 at the cellular level would be appreciated. Ideally this would involve combined calcium imaging and electrophysiological studies demonstrating intracellular calcium dynamics following K+ channel modulation. However, given the difficulties in this executing this technique, simple calcium imaging studies would still provide mechanistic evidence to strengthen the isometric tension studies.

Minor:

Terminology such as “the intracellular Ca2+ stores were removed” should be avoided. Granted, CPA treatment mediates store depletion, but it does not remove the stores per se. 

Reviewer 3 Report

Reviewer comments:

The article of Zak et al, is a very interesting study and welcomed to the scientific community, showing the importance of oxytocin on KCa channel feedback that facilitates CyPPA suppression of uterine contractions. However, there are some critical points that should be clarified before acceptance for publication. Please see the major and minor comments bellow:

- ''...strips with 30 μM Gd3+ to block NSCC influx.'' - to state again Gd3+

- ''Small conductance Ca2+-activated potassium channels (KCa2.3) have been identified as promising targets to attenuate premature uterine contractions'' - probably better if the authors don't state the specific member KCa2.3 but just to leave KCa channels. If the authors want, they can state another fact about KCa2.3.

- I would suggest to the authors to enrich the introduction with more info about oxytocin.

- Otherwise, I appreciate a lot the Introduction and Discussion sections.

- On the other hand the Results and Methods sections they need major revisions.

- The action of CyPPA on KCa is already known.

- Apamin is not a specific blocker for KCa2.3.

- '' These effects were most evident in contraction amplitude and AUC, while contraction frequency was not significantly different'' - in fig 1A upper panel, between Oxy 0.1uM and Oxy 1uM there are almost 19 spikes whereas in the down panel there are 7. The authors need better representations. Both panels should have the same OY axes.

- Redundant to add AUC (area under curve) in right panel of fig 1B.

- How the authors prepared the oxytocin stock solution?

- How the authors prepared the CyPPA stock solution?

- How the authors prepared the Apamin stock solution?

- A very important point all through the manuscript. In figures, the analysis is poorly described. Beside the fact that authors should be clear about n=strips, n=animals, they should also show numerical values of the statistics and the software used to do the comparison. I would suggest an extended data table for statistical analysis containing the figure #, normality test, type of statistical test, n value and what it represents, mean value, SEM value, p value, Cohen's effect-size value, ANOVA values

- ''...with 10 μM CPA before adding 100 nM oxytocin.'' - to be stated again what is CPA#

- ''... uterine contractions, the intracellular Ca2+ stores were removed...''; ''... depletion of internal Ca2+ stores with cyclopiazonic acid...'' - is CPA used for depletion/removal or for inhibition?

- ''... contractions compared to vehicle (DMSO; Figure 2)'' - redundant to add DMSO

- in fig 2A both panels to have the same OY

- in fig 2B both panels to have the same OY

- why did authors chose the Oxy 0.1uM concentration, in fig2?

- what is the comparison of amplitude, frequency and AUC between DMSO and CPA, in fig 2?

- what is the comparison of amplitude, frequency and AUC between Oxy 0.1uM after DMSO and CPA, in fig 2?

- in fig2B the values are counted after the treatment with CyPPA concentrations? Then, is no difference between frequency??? Definitely, you need other representations

- How the authors prepared the Gd3+ stock solution?

- in fig 3A both panels to have the same OY

- in fig 3B both panels to have the same OY

- the traces in fig3 are so different from those in fig2. Can authors explain?

- in fig 3A lower panel, between CyPPA 10uM and 30uM, did authors count the very small amplitude events? What are they?

- ''2.4. KCa2.3 channels and IP3Rs exhibit overlapping punctate expression patterns in the myometrium.'' - with a dot or not?

- ''...myometrial smooth muscle (Fig-ure 4). There was considerable overlap in the staining patterns...'' - orthography

- ''Lower right panel shows distinct areas where...'' - I advice authors to make a different counting or representation of the images. How many strips? How many images? How many animals?

- Please, it is very useful for the entire community if we can all agree with the RRID - Research Resource Identifiers or to state at least the reference number of all other reagents, antibodies, software, equipment.

- ''...containing in mM: 119 NaCl, 4.7 KCl, 1.2 MgSO4, 2.0 CaCl2, 23 NaHCO3, 10.5 glucose, 0.026 EDTA, 1.2 KH2PO4...'' - what is the pH and osmolarity?

- ''...in 4C physiological and at 37O C aerated...'' - the degree is different represented

- ''...at 4OC or IP3R primary...'' - is ''or'' or ''and''???

Round 2

Reviewer 1 Report

The author discussed my points in the discussion. There are some minor corrections, such as the lack of scale fluolescent images in different sizes, but I understand the argument and have no further comments.

Author Response

We thank the Reviewer for the helpful and supportive comments.

Reviewer 2 Report

Satisfactory response to comments.

Author Response

We thank the Reviewer for the supportive comments.

Reviewer 3 Report

Reviewer comments:

The article of Zak et al, is on a good track to get published but still they have to be more transparent about their work.

Comment: '' These effects were most evident in contraction amplitude and AUC, while contraction frequency was not significantly different'' -in fig 1A upper panel, between Oxy 0.1uM and Oxy 1uM there are almost 19 spikes whereas in the down panel there are 7. The authors need better representations. Both panels should have the same OY axes. 20. the traces in fig3 are so different from those in fig2. Can authors explain?

Response: It is important to note that spontaneous contractions and oxytocin effects are variable from preparation to preparation. This is a general feature we have observed in rodent as well as more recently in human uterine strips (in preparation). Different sampled inherent pacemakers as well as general orientation of muscle relative to the force transducer pins can produce a range of frequencies and amplitudes among tissues. Nevertheless, for each tissue, consistent and persistent contractile patterns exist, allowing for quantification over the time intervals assessed. Also, note that assessment of AUC provides a global quantification of force over time, separate from the specific effects on frequency or amplitude. In order to account for the inherent variability among tissues, we assess the effect a perturbation has on any parameter (e.g. frequency) relative to the status of that parameter before the perturbation, and express this as a relative change. In the case of frequency in Fig 1, note that despite the brief increase in frequency due to the first oxytocin addition, over the course of successive exposures, there is actually only a marginal notable change, if any (i.e. at end of 1 uM oxytocin there are ~5 contractions over 10 min which is comparable to the frequency before any oxytocin was added). Overall, there was a slight general trend of increasing frequency after oxytocin exposure. However, the central question was whether the frequencies differed in CYPPA treated and untreated preparations. They were not statistically different. Regarding the axes, again, the point was not to make direct comparisons of absolute amplitudes but rather to assess relative change with perturbation. In order to avoid unnecessary white space and compression of the recording, we feel it is justified to fit the axes to the data. We have expanded the Methods and Results sections to more clearly explain our rationale, approach and findings.

New comment: I thank the authors for the response. Still, even more because the authors say that spontaneous contractions and oxytocin effects are variable from preparation to preparation, I would like to see other representative traces, especially for fig 2 and 3. In fig2C and fig3C the frequency is expressing the existence of spikes whereas in the traces the contractions are missing.

Comment: A very important point all through the manuscript. In figures, the analysis is poorly described. Beside the fact that authors should be clear about n=strips, n=animals, they should also show numerical values of the statistics and the software used to do the comparison. I would suggest an extended data table for statistical analysis containing the figure #, normality test, type of statistical test, n value and what it represents, mean value, SEM value, p value, Cohen's effect-size value, ANOVA values

Response: For all experiments, n=number of animals. Analysis was performed using GraphPad Prism 9. All data sets were assessed for normality using Shapiro-Wilk test. Two-way ANOVA was applied for comparisons of interest i.e., two independent variables. Unpaired t-test was used for new two-variable data assessing direct CPA and Gd3+ effects. Alpha was 0.05 in all cases. The figures were designed to provide data without redundancy or omission; we have included relevant p values where informative.

New comment: I don't see any mean value, SEM value etc. I enforce the need of an extended data table for stats.

Comment: in fig 2A both panels to have the same OY …in fig 2B both panels to have the same OY 19. in fig 3A both panels to have the same OY… in fig 3B both panels to have the same OY

Response: As noted above, in this case, since we are not directly comparing absolute contraction amplitudes, we prefer to fit the axes to the data and prevent compression of the data plots.

New comment: All fig2C and fig3C, should have the same OY, from 0 to 100, from 20 to 20 with no minor lines.

Comment: what is the comparison of amplitude, frequency and AUC between DMSO and CPA, in fig 2?

Response: Based on the Reviewer’s request, we have added bar graphs showing the relative effects of CPA and Gd3+ in Figures 2 and 3, respectively.

Comment: what is the comparison of amplitude, frequency and AUC between Oxy 0.1uM after DMSO and CPA, in fig 2?

Response: Since the focus of this protocol was not to assess oxytocin pretreatment but rather to quantify the CyPPA impact under the pretreatment scenario, we feel quantification of this pretreatment would not add significantly to the value of the manuscript.

New comment: because fig2B is showing some differences then, the results in fig2C should be commented with precaution. Because of this and because the authors did not analysed '' Comment: what is the comparison of amplitude, frequency and AUC between Oxy 0.1uM after DMSO and CPA, in fig 2?''  I would then suggest to keep out from the main manuscript the fig2B and fig3B, or add them as supplementary

Comment: in fig2B the values are counted after the treatment with CyPPA concentrations? Then, is no difference between frequency???Definitely, you need other representations

Response: As alluded to earlier, there is some variability among preparations, but the recordings shown are in fact representative examples of the experiments. In some cases, amplitudes may be reduced by

CyPPA with no impact on frequency or events may be prevented entirely, resulting in a net frequency of zero. The overall difference with and without CPA was the focus in Fig 2B; here the p value for frequency is 0.0822. This was not considered significant for the alpha value (0.05) chosen. We have now added pertinent p values for the figures in order to clarify.

New comment: I am sorry, even more because the authors say that there is some variability among preparations, I would like to see other representative traces

Comment: Please, it is very useful for the entire community if we can all agree with the RRID - Research Resource Identifiers or to state at least the reference number of all other reagents, antibodies, software, equipment.

Response: We have included this information in the revised manuscript.

New comment: The authors did not fully answered the comment. If someone need to duplicate your study, they need:

Reagent/equipment/software

Source (Company, City, State, Country)

Catalogue Number
